# Goal-directed Generation of Discrete Structures with Conditional Generative Models

**Amina Mollaysa**
University of Geneva
University of Applied Sciences Western Switzerland
`maolaaisha.aminanmu@hesge.ch`

**Brooks Paige**
University College London
Alan Turing Institute
`b.paige@ucl.ac.uk`

**Alexandros Kalousis**
University of Geneva
University of Applied Sciences Western Switzerland
`alexandros.kalousis@hesge.ch`

## Abstract

Despite recent advances, goal-directed generation of structured discrete data remains challenging. For problems such as program synthesis (generating source code) and materials design (generating molecules), finding examples which satisfy desired constraints or exhibit desired properties is difficult. In practice, expensive heuristic search or reinforcement learning algorithms are often employed. In this paper we investigate the use of conditional generative models which directly attack this inverse problem, by modeling the distribution of discrete structures given properties of interest. Unfortunately, maximum likelihood training of such models often fails with the samples from the generative model inadequately respecting the input properties. To address this, we introduce a novel approach to directly optimize a reinforcement learning objective, maximizing an expected reward. We avoid high-variance score-function estimators that would otherwise be required by sampling from an approximation to the normalized rewards, allowing simple Monte Carlo estimation of model gradients. We test our methodology on two tasks: generating molecules with user-defined properties, and identifying short python expressions which evaluate to a given target value. In both cases we find improvements over maximum likelihood estimation and other baselines.

## 1 Introduction

Machine learning models for structured data such as program source code and molecules typically represent the data as a sequence of discrete values. However, even as recent advances in sequence models have made great strides in predicting properties from the sequences, the inverse problem of generating sequences for a given set of pre-defined properties remains challenging. These sorts of structure design problems have great potential in many application areas; e.g., in materials design, where one may be interested in generating molecular structures appropriate as batteries, photovoltaics, or drug targets. However, as the underlying sequence is discrete, directly optimizing it with respect to target properties becomes problematic: gradients are unavailable, meaning we can not directly estimate the change in (say) a material that would correspond to a desired change in properties.

An open question is how well machine learning models can help us quickly and easily explore the space of discrete structures that correspond to particular properties. There are a few approaches based on generative models which aim to directly simulate likely candidates. Neural sequence models [23, 2] are used to directly learn the conditional distributions by maximizing conditional log likelihoods and have shown potential in text generation, image captioning and machine translation.

Fundamentally, a maximum likelihood (ML) objective is a poor choice if our main goal is not learning the exact conditional distribution over the data, but rather to produce a diverse set of generations which match a target property. Reinforcement learning (RL) offers an alternative, explicitly aiming to maximize an expected reward. The fundamental difference is between "many to one" and "one to many" settings. A ML objective works well for prediction tasks where the goal is to match each input to its exact output. For instance, consider the models which predict properties from molecules, since each molecule has a well-determined property. However, for the inverse problem the data severely underspecifies the mapping, since any given property combination has many diverse molecules which match; nearly all of these are not present in the training data. As the ML objective is only measured on the training pairs, any output that is different from the training data target is penalized. Therefore, a model which generates novel molecules with the correct properties would be penalized by ML training, as it does not produce the exact training pairs; but, it would have a high reward and thus be encouraged by an RL objective.

Recent work has successfully applied policy gradient optimization [25] on goal oriented sequence generation tasks [6, 11]. These methods can deal with the non-differentiability of both the data and the reward function. However, policy gradient optimization becomes problematic especially in high dimensional action spaces, requiring complicated additional variance reduction techniques [10]. Those two schemes, ML estimation and RL training, are both equivalents to minimizing a KL divergence between an exponentiated reward distribution (defined accordingly) and model distribution but in the opposite directions [18]. If all we care about is generating realistic samples, the direction corresponding to the RL objective is the more appropriate to optimize [13]: the only question is how to make training efficient.

We formulate the conditional generation of discrete sequence as a reinforcement learning objective to encourage the generation of sequences with specific desired properties. To avoid the inefficiency of the policy gradient optimization, which requires large sample sizes and variance reduction techniques, we propose an easy alternative formulation that instead leverages the underlying similarity structure of a training dataset.

## 2 Methods

In this section, we introduce a new approach which targets an RL objective, maximizing expected reward, to directly learn conditional generative models of sequences given target properties. We sidestep the high variance of the gradient estimation which would arise if we directly apply policy gradient methods, by an alternative approximation to the expected reward that removes the need to sample from the model itself.

Suppose we are given a training set of pairs $\mathcal{D} = \{(\mathbf{x}_i, \mathbf{y}_i)\}, i = 1, \ldots, N$, where $\mathbf{x}_i = x_{i,1\ldots T}$ is a length $T$ sequence of discrete variables, and $\mathbf{y}_i \in \mathcal{Y}$ is the corresponding output vector, whose domain may include discrete and continuous dimensions. Assume the input-output instances $(\mathbf{x}_i, \mathbf{y}_i)$ are i.i.d. sampled from some unknown probability distribution $\tilde{p}(\mathbf{x}, \mathbf{y})$, with $f$ the ground truth function that maps $\mathbf{x}_i$ to $\mathbf{y}_i$ for all pairs $(\mathbf{x}_i, \mathbf{y}_i) \sim \tilde{p}(\mathbf{x}, \mathbf{y})$. Our goal is to learn a distribution $p_\theta(\mathbf{x}|\mathbf{y})$ such that for a given $\mathbf{y}$ in $\tilde{p}(\mathbf{y})$, we can generate samples $\mathbf{x}$ that have properties close to $\mathbf{y}$.

### 2.1 Maximizing expected reward

We can formulate this learning problem in a reinforcement learning setting, where the model we want to learn resembles learning a stochastic policy $p_\theta(\mathbf{x}|\mathbf{y})$ that defines a distribution over $\mathbf{x}$ for a given state $\mathbf{y}$. For each $\mathbf{x}$ that is generate for a given state $\mathbf{y}$, we can define a reward function $R(\mathbf{x}; \mathbf{y})$ such that it assigns higher value for $\mathbf{x}$ when $d(f(\mathbf{x}), \mathbf{y})$ is small and vice versa, where $d$ is a meaningful distance defined on $\mathcal{Y}$. This model can be learned by maximizing the expected reward

$$\mathcal{J} = \mathbb{E}_{\tilde{p}(\mathbf{y})} \mathbb{E}_{p_\theta(\mathbf{x}|\mathbf{y})} [R(\mathbf{x}; \mathbf{y})]. \tag{1}$$

When there is a natural notion of distance $d(\mathbf{y}, \mathbf{y}')$ for values in $\mathcal{Y}$, then we can define a reward $R(\mathbf{x}; \mathbf{y}) = \exp\{-\lambda d(f(\mathbf{x}), \mathbf{y})\}$. If the model $p_\theta(\mathbf{x}|\mathbf{y})$ defines a distribution over discrete random variables or if the reward function is non-differentiable, a direct approach requires admitting high-variance score-function estimators of the gradient of the form

$$\nabla_\theta \mathcal{J} = \mathbb{E}_{\tilde{p}(\mathbf{y})} \mathbb{E}_{p_\theta(\mathbf{x}|\mathbf{y})} [R(\mathbf{x}; \mathbf{y}) \nabla_\theta \log p_\theta(\mathbf{x}|\mathbf{y})]. \tag{2}$$

The inner expectation in this gradient estimator would typically be Monte Carlo approximated via sampling from the model $p_\theta(\mathbf{x}|\mathbf{y})$. The reward is often sparse in high dimensional action spaces,

leading to a noisy gradient estimate. Typically, to make optimization work with score-function gradient estimators, we need large samples sizes, control variate schemes, or even warm-starting from pre-trained models. Instead of sampling from the model distribution and look for the direction where we get high rewards, we consider an alternative breakdown of the objective to avoid direct Monte Carlo simulation from $p_\theta(\mathbf{x}|\mathbf{y})$.

Assume we have a finite non-negative reward function $R(\mathbf{x}; \mathbf{y})$, with $0 \leq R(\mathbf{x}; \mathbf{y}) < \infty$, and let $c(\mathbf{y}) = \sum_{\mathbf{x}} R(\mathbf{x}; \mathbf{y})$. We can then rewrite the objective in Eq. (1), using the observation that

$$\mathbb{E}_{p_\theta(\mathbf{x}|\mathbf{y})}[R(\mathbf{x}; \mathbf{y})] = c(\mathbf{y})\mathbb{E}_{\bar{R}(\mathbf{x}|\mathbf{y})}[p_\theta(\mathbf{x}|\mathbf{y})], \tag{3}$$

where we take the expectation instead over the "normalized" rewards distribution $\bar{R}(\mathbf{x}|\mathbf{y}) = R(\mathbf{x}; \mathbf{y})/c(\mathbf{y})$. (A detailed derivation is in Appendix A.) Leaving aside for the moment any practical challenges of normalizing the rewards, we note that this formulation allows us instead to employ a path-wise derivative estimator [21]. Using the data distribution to approximate expectations over $\tilde{p}(\mathbf{y})$ in Eq. 2, we have

$$\nabla_\theta \mathcal{J} \approx \frac{1}{N} \sum_{i=1}^{N} c(\mathbf{y}_i)\mathbb{E}_{\bar{R}(\mathbf{x}|\mathbf{y}_i)}[\nabla_\theta p_\theta(\mathbf{x}|\mathbf{y}_i)]. \tag{4}$$

To avoid numeric instability as $p_\theta(\mathbf{x}|\mathbf{y})$ may take very small values, we instead work in terms of log probabilities. This requires first noting that $\arg\max_\theta \mathcal{J} = \arg\max_\theta \log \mathcal{J}$ where we then have:

$$\log \mathcal{J} = \log \left( \mathbb{E}_{\tilde{p}(\mathbf{y})} c(\mathbf{y})\mathbb{E}_{\bar{R}(\mathbf{x}|\mathbf{y})}[p_\theta(\mathbf{x}|\mathbf{y}_i)] \right) \geq \mathbb{E}_{\tilde{p}(\mathbf{y})} \log \left( c(\mathbf{y})\mathbb{E}_{\bar{R}(\mathbf{x}|\mathbf{y})}[p_\theta(\mathbf{x}|\mathbf{y}_i)] \right)$$
$$= \mathbb{E}_{\tilde{p}(\mathbf{y})} \log \mathbb{E}_{\bar{R}(\mathbf{x}|\mathbf{y})}[p_\theta(\mathbf{x}|\mathbf{y})] + const.$$

From Jensen's inequality, we have $\log \mathbb{E}_{\bar{R}(\mathbf{x}|\mathbf{y})}[p_\theta(\mathbf{x}|\mathbf{y})] \geq \mathbb{E}_{\bar{R}(\mathbf{x}|\mathbf{y})}[\log p_\theta(\mathbf{x}|\mathbf{y})]$, which motivates optimizing instead a lower-bound on $\log \mathcal{J}$, an objective we refer to as

$$\mathcal{L} = \mathbb{E}_{\tilde{p}(\mathbf{y})}\mathbb{E}_{\bar{R}(\mathbf{x}|\mathbf{y})}[\log p_\theta(\mathbf{x}|\mathbf{y})] \tag{5}$$

Again using the data distribution to approximate expectations over $\tilde{p}(\mathbf{y})$, the gradient is simply

$$\nabla_\theta \mathcal{L} \approx \frac{1}{N} \sum_{i=1}^{N} \mathbb{E}_{\bar{R}(\mathbf{x}|\mathbf{y}_i)}[\nabla_\theta \log p_\theta(\mathbf{x}|\mathbf{y}_i)]. \tag{6}$$

## 2.2 Approximating expectations under the normalized reward distribution

Of course, in doing this we have introduced the potentially difficult task of sampling directly from the normalized reward distribution. Fortunately, if we are provided with a reasonable training dataset $\mathcal{D}$ which well represents $\tilde{p}(\mathbf{x}, \mathbf{y})$ then we propose not sampling new values at all, but instead re-weighting examples $\mathbf{x}$ from the dataset as appropriate to approximate the expectation with respect to the normalized reward.

For a fixed reward function $R(\mathbf{x}, \mathbf{y}) = \exp\{-\lambda d(f(\mathbf{x}), \mathbf{y})\}$, when restricting to the training set we can instead re-express the normalized reward distribution in terms of a distribution over training indices. Given a fixed $\mathbf{y}_i$, consider the rewards as restricted to values $\mathbf{x} \in \{\mathbf{x}_1, \ldots, \mathbf{x}_N\}$; each potential $\mathbf{x}_j$ has a paired value $\mathbf{y}_j = f(\mathbf{x}_j)$. Using our existing dataset to approximate the expectations in our objective, we have for each $\mathbf{y}_i$

$$\mathbb{E}_{\bar{R}(\mathbf{x}|\mathbf{y}_i)}[\log p_\theta(\mathbf{x}|\mathbf{y}_i)] \approx \sum_{j=1}^{N} \bar{R}(\mathbf{x}_j|\mathbf{y}_i) \log p_\theta(\mathbf{x}_j|\mathbf{y}_i) \approx \mathbb{E}_{p(j|i)}[\log p_\theta(\mathbf{x}_j|\mathbf{y}_i)] \tag{7}$$

where the distribution over indices $p(j|i)$ is defined as

$$p(j|i) = \frac{R(\mathbf{x}_j; \mathbf{y}_i)}{\sum_{j=1}^{N} R(\mathbf{x}_j; \mathbf{y}_i)}. \tag{8}$$

Note that there are two normalized distributions which we consider. The first $\bar{R}(\mathbf{x}|\mathbf{y}_i)$ is defined by normalizing the reward across the entire space of possible values $\mathbf{x}$,

$$\bar{R}(\mathbf{x}|\mathbf{y}_i) = \frac{R(\mathbf{x}; \mathbf{y}_i)}{\sum_{\mathbf{x}} R(\mathbf{x}; \mathbf{y}_i)} = \frac{R(\mathbf{x}; \mathbf{y}_i)}{c(\mathbf{y}_i)} \tag{9}$$

The second is $p(j|i)$ which is defined by normalizing instead across the empirical distribution of values $\mathbf{x}$ in the training set, yielding the distribution over indices as given in equation 8. The restriction of $\bar{R}(\mathbf{x}|\mathbf{y}_i)$ to the discrete points in the training set is not the same and our approximation is off by a scalar multiplicative factor $\sum_{j=1}^{N} R(\mathbf{x}_j|\mathbf{y}_i)/\sum_{\mathbf{x}} R(\mathbf{x}|\mathbf{y}_i)$. However, this factor does not dramatically affect outcomes: its value is *independent of* $\mathbf{x}$, depending only on $\mathbf{y}_i$. Any bias which is introduced would come into play only in the evaluation of Eq. (6), where entries from different $\mathbf{y}$ values may effectively be assigned different weights. So while this may affect unconditional generations from the model, and means that some $\mathbf{y}$ values may be inappropriately considered "more important" than others during training, it should not directly bias *conditional* generation for a fixed $\mathbf{y}$. Full discussion of this point is in Appendix B.

Thus, for any choice of reward, we can define a sampling distribution from which to propose examples $\mathbf{x}_j$ given $\mathbf{y}_i$ by computing and normalizing the rewards across $(i, j)$ pairs in the dataset. While naïvely computing this sampling distribution for every $i$ requires constructing an $N \times N$ matrix, in practice typically very few of the $N$ datapoints have normalized rewards which are numerically much greater than zero, allowing easy pre-computation of a $N$ arrray of dimention $K_i$ where $K$ is the maximum number of non-zero elements each entry $i$, under the distance $d(f(\mathbf{x}_j), \mathbf{y}_i)$. This can then later be used as a sampling distribution for $\mathbf{x}_j$ at training time.

## 2.3 Sequence diversification

Our sampling procedure operates exclusively over the training instances. This, coupled with the fact that we operate over sequences of discrete elements, can restrict in a considerable manner the generation abilities of our model. To encourage a more diverse exploration of the sequence space and thus more diverse generations we couple our objective with an entropy regulariser. In addition to the likelihood of the generated sequences, we propose also maximizing their entropy, with

$$\max_{\theta} \sum_{i=1}^{N} [\mathbb{E}_{\bar{R}(\mathbf{x}|\mathbf{y}_i)}[\log p_\theta(\mathbf{x}|\mathbf{y}_i)] + \lambda H(p_\theta(\mathbf{x}|\mathbf{y}_i))]. \tag{10}$$

In a discrete sequence model the gradient of the entropy term can be computed as

$$\nabla_\theta H(p_\theta(\mathbf{x}|\mathbf{y})) = -\nabla_\theta \mathbb{E}_{p_\theta(\mathbf{x}|\mathbf{y})} \log p_\theta(\mathbf{x}|\mathbf{y}) = -\mathbb{E}_{p_\theta(\mathbf{x}|\mathbf{y})}[(1 + \log p_\theta(\mathbf{x}|\mathbf{y}))\nabla_\theta \log p_\theta(\mathbf{x}|\mathbf{y})]; \tag{11}$$

for details see Appendix C. Suffice it to say, a naïve Monte Carlo approximation suffers from high variance and sample inefficiency. We instead use an easily-differentiable approximation to the entropy. We decompose the entropy of the generated sequence into a sequence of individual entropy terms

$$H(p_\theta(\mathbf{x}|\mathbf{y})) = -\mathbb{E}_{p_\theta(\mathbf{x}|\mathbf{y})} \sum_{t=1}^{T} \log p_\theta(\mathbf{x}_t|\mathbf{x}_{1:t-1}, \mathbf{y}) = H(p_\theta(\mathbf{x}_1|\mathbf{y})) + \sum_{t=2}^{T} E_{p_\theta(\mathbf{x}_{1:t-1}|\mathbf{y})}[H(p_\theta(\mathbf{x}_t|\mathbf{x}_{1:t-1}, \mathbf{y}))].$$

We can compute analytically the entropy of each individual $p_\theta(\mathbf{x}_t|\mathbf{x}_{1:t-1}, \mathbf{y})$ term, since this is a discrete probability distribution with a small number of possible outcomes given by the dictionary size, and use sampling to generate the values we condition on at each step. Instead of using a Monte Carlo estimation of the expectation term, we generate the sequence in a greedy manner, selecting the maximum probability element, $\mathbf{x}_t^*$, at each step. This results in approximating the entropy term as

$$H(p_\theta(\mathbf{x}|\mathbf{y})) \approx H[p_\theta(\mathbf{x}_1|\mathbf{y})] + \sum_{t=2}^{T} \left[ H[p_\theta(\mathbf{x}_t|\mathbf{x}_{1:t-1}^*, \mathbf{y})] \right].$$

Its gradient is straightforward to compute since each individual entropy term can be computed analytically. In Appendix J we provide an experimental evaluation suggesting the above approximation outperforms simple Monte Carlo based estimates, as well as the straight-through estimator.

## 3 Experiments

We evaluate our training procedure on two conditional discrete structure generation tasks: generating python integer expressions that evaluate to a given value, and generating molecules which should exhibit a given set of properties. We do a complete study of our model without the use of the entropy-based regulariser and then explore the behavior of the regulariser.

```
S —> Expr Op Expr [1.0]
Expr —> Number [0.4] | Expr Op Expr [0.4] |
        L Expr Op Expr R [0.2]
Number —> Nonzero Digits [0.9] | Nonzero [0.1]
Nonzero —> 1 [0.1111] | 2 [0.1111] | 3 [0.1111] |
           4 [0.1111] | 5 [0.1111] | 6 [0.1111] |
           7 [0.1111] | 8 [0.1111] | 9 [0.1111]
Digits —> Digit [0.95] | Digit Digits [0.05]
Digit —> 0 [.1] | Nonzero [0.9]
Op —> '+' [0.3] | '−' [0.3] |
      '*' [0.2] | '//' [0.2]
L —> '(' [1.0]
R —> ')' [1.0]
```

Listing 1: CFG for inverse calculator

| Objective | Valid | Unique | Novel |
|---|---|---|---|
| ML | $0.9888 \pm 0.0002$ | $0.9681 \pm 0.0004$ | $0.9301 \pm 0.0003$ |
| Ours | $0.9903 \pm 0.0003$ | $0.9635 \pm 0.0006$ | $0.9271 \pm 0.0005$ |

Table 1: Python integer expression generation results

| | MAE | Accuracy | Within $\pm 3$ | $-\log p(\mathbf{x}|\mathbf{y})$ |
|---|---|---|---|---|
| ML | $13.917 \pm 0.117$ | $\mathbf{0.166 \pm 0.001}$ | $0.596 \pm 0.001$ | $\mathbf{1.830}$ |
| Ours | $\mathbf{11.823 \pm 0.145}$ | $\mathbf{0.166 \pm 0.001}$ | $\mathbf{0.682 \pm 0.001}$ | $1.986$ |

Table 2: Python integer expression *conditional* generation results

In all experiments, we model the conditional distributions $p_\theta(\mathbf{x}|\mathbf{y})$ using a 3-layer stacked LSTM sequence model; for architecture details see Appendix D and Fig. D.3. We evaluate the performance of our model both in terms of the quality of the generations, as well as its sensitivity to values of the conditioning variable $\mathbf{y}$. Generation quality is measured by the percentage of valid, unique, and novel sequences that the model produces, while the quality of the conditional generations is evaluated by computing the error between the value $\mathbf{y}$ we condition on, and the property value $f(\mathbf{x})$ that the generated sequence $\mathbf{x}$ exhibits. We compared our model against a vanilla maximum likelihood (ML) trained model, where we learn the distribution $p_\theta(\mathbf{x}|\mathbf{y})$ by maximizing the conditional log-likelihood of the training set. We also test against two additional data augmentation baselines, including the reward augmented maximum likelihood (RAML) procedure of Norouzi et al. [18].

### 3.1 Conditional generation of mathematical expressions

Before considering the molecule generation task, we first look at a simpler constrained "inverse calculator" problem, in which we attempt to generate python integer expressions that evaluate to a certain target value. We generate synthetic training data by sampling form the probabilistic context free grammar presented in Listing 1. We generate approximately 300k unique, valid expressions of length 30 characters or less, that evaluate to integers in the range $(-1000, 1000)$; we split them into training, test, and validation subsets. Full generation details are in Appendix E.

We define a task-specific reward based on permitting a small squared-error difference between the equations' values and the conditioning value; i.e. $R(\mathbf{x}|y) = \exp\left\{-\frac{1}{2}(f(\mathbf{x}) - y)^2\right\}$, which means for a given candidate $y_i$ the normalized rewards distribution forms a discretized normal distribution. We can sample appropriate matching expressions $\mathbf{x}_j$ by first sampling a target value from $y'$ from a Gaussian $\mathcal{N}(y'|y,)$ truncated to $(-999, 999)$, rounding the sampled $y'$, and then uniformly sampling $\mathbf{x}$ from all training examples such that $\text{EVAL}(\mathbf{x}) = \text{round}(y')$. We found training to be most stable by pre-sampling these, drawing 10 values of $\mathbf{x}$ for each $y_i$ in the training and validation sets and storing these as a single, larger training and validation set which can be used to directly estimate the expected reward.

To evaluate the performance of the generated integer expressions, we sample 25 expressions for 10k values in the test set; we repeat this process 6 times and report means and standard deviations. Generated expressions which evaluate to non-finite values, or to values outside the $(-1000, 1000)$ range, are discarded as invalid. In Table 1 we show the statistics for the validity, uniqueness, and novelty (relative to the training set) of the generated expressions.s Both models perform well, and quite comparably; about 99% of generated expressions are valid, with few duplicates either of the other samples or of the training expressions. Table 2 reports (for the valid expressions) the mean absolute error (MAE), the exact accuracy (i.e. whether $\text{EVAL}(\mathbf{x}) = y$), an "approximate" accuracy checking if the value is within $\pm 3$, as well as the negative log likelihood on the test set. While the exact accuracy does not change, we do see better performance in our training regime at both MAE and approximate accuracy, suggesting that "wrong" expressions are in some sense less wrong. We also note the tradeoff between objectives: our model trained to improve generation accuracy slightly underfits in terms of test log likelihood.

### 3.2 Conditional generation of molecules

We now attempt to learn a model that can directly generate molecules that exhibit a given set of properties. We experiment with two datasets: QM9 [19] which contains 133k organic compounds that have up to nine heavy atoms and ChEMBL [17] of molecules that have been synthesized and tested against biological targets, which includes relatively larger molecules (up to 16 heavy atoms). For details of the datasets see Appendix D. We represent molecules by SMILES strings [24], and

| | QM9 | | | ChEMBL | | |
|---|---|---|---|---|---|---|
| Model | Valid | Unique | Novel | Valid | Unique | Novel |
| ML | 0.962 | **0.967** | **0.366** | 0.895 | **0.999** | **0.990** |
| Ours | **0.989** | 0.963 | 0.261 | **0.945** | 0.9986 | 0.981 |

Table 3: Molecule generation quality

| | QM9 | | |
|---|---|---|---|
| Model | Validity | Unicity | Novelty |
| Classic data augmentation | 0.937 | 0.980 | 0.464 |
| RAML-like data augmentation | 0.940 | **0.986** | **0.747** |
| Ours + Entropy regularizer ($\lambda = 0.0008$) | **0.977** | 0.961 | 0.274 |

Table 4: Molecule generation quality

| Model | rotatable bonds | aromatic rings | logP | QED | TPSA | bertz | mol weight | fluorine count | # rings |
|---|---|---|---|---|---|---|---|---|---|
| | | | | QM9: MSE | | | | | |
| ML | 0.0468 | 0.0014 | 0.0390 | 0.0010 | 11.18 | 80.77 | 4.425 | 0.0023 | 0.0484 |
| Ours | **0.0166** | **0.0005** | **0.0184** | **0.0004** | **3.859** | **63.67** | **1.184** | **0.0004** | **0.0120** |
| | | | | QM9: Correlation coefficient | | | | | |
| ML | 0.9809 | 0.9944 | 0.9805 | 0.9063 | 0.9871 | **0.9843** | 0.9651 | 0.9783 | 0.9817 |
| Ours | **0.9937** | **0.9972** | **0.9901** | **0.9634** | **0.9954** | 0.9840 | **0.9887** | **1.0000** | **0.9948** |
| | | | | ChEMBL: MSE | | | | | |
| ML | **0.1552** | 0.0388 | 0.1450 | 0.0050 | **27.64** | **1708.** | 103.9 | 0.0128 | 0.0226 |
| Ours | 0.1555 | **0.0268** | **0.1320** | **0.0046** | 35.05 | 2512. | 174.9 | **0.0074** | **0.0191** |
| | | | | CheEMBL: Correlation coefficient | | | | | |
| ML | **0.9936** | 0.9862 | 0.9777 | 0.9450 | **0.9906** | **0.9934** | **0.9956** | 0.9940 | 0.9931 |
| Ours | 0.9934 | **0.9901** | **0.9796** | **0.9496** | 0.9878 | 0.9902 | 0.9926 | **0.9966** | **0.9943** |

Table 5: Conditional generation performance for the molecules datasets.

condition on nine molecule properties which are present in the goal-directed generation tasks of Brown et al. [5]: number of rotatable bonds, number of aromatic rings, logP, QED score, tpsa, bertz, molecule weight, atom counter and number of rings; these properties can all be readily estimated by open-source chemoinformatics software RDKit (`www.rdkit.org`). For training purposes we normalize the properties values to a zero mean and a standard deviation of one.

**Sampling from the reward-based distribution**  During training, we need samples from $\bar{R}(\mathbf{x}|\mathbf{y}_i)$ for each $\mathbf{y}_i$ from the training set. We define the reward as

$$R(\mathbf{x}; \mathbf{y}_i) = \begin{cases} \exp(-\lambda \ell_1(f(\mathbf{x}), \mathbf{y}_i)), & \text{if } \ell_1(f(\mathbf{x}), \mathbf{y}_i) \leq \epsilon \\ 0, & \text{otherwise}, \end{cases} \tag{12}$$

where $\ell_1(\cdot, \cdot)$ is the $\ell_1$ distance of its arguments, and $\lambda$ is a temperature hyper-parameter that prevents $p(\cdot|i)$, i.e., the approximation of $\bar{R}(\mathbf{x}; \mathbf{y}_i)$ using the training set, from becoming uniform over the non-zero reward samples; it controls how peaked is the $p(\cdot|i)$ distribution at the ground truth. We determine the values of the hyper-parameters based on the statistics of the $\ell_1(f(\mathbf{x}_j), \mathbf{y}_i)$ distance on the training set, details are explained section D. We pre-compute the probability table $p(\cdot|i)$ for all $\mathbf{y}_i$ in the training set and save the non-zero entries and the corresponding $\mathbf{x}$ indices. During training, for each $\mathbf{y}_i$ in the mini batch, we sample ten $\mathbf{x}$ samples from $p(\cdot|i)$.

**Results**  To evaluate the performance we do one sample generation for each $\mathbf{y}_i$ in the test set from the learned model $p_\theta(\mathbf{x}|\mathbf{y}_i)$. In Table 3, we provide the generation performance results over full test set. In the Table 5, we present the result for conditional generation where we do one sample generation per $\mathbf{y}_i$ and calculate the error between the obtained molecules property and target $\mathbf{y}_i$. To account for the randomness that occur due to the sampling, we repeat such process ten times and report the statistics over ten trials. In the case of ChEMBL, repeating such generation ten times over a test set of size 238k was too expensive so we limited to 10k randomly selected instances. In Appendix G we report the results of a single molecule sampling over the full test set, consistent with the results we provide here.

For QM9 our model has smaller MSE than the ML baseline model and better correlation on all properties. In the larger ChEMBL dataset the results are mixed, with our model achieving an MSE lower than that of the ML in five out of the nine properties and higher in the remaining four. More results are presented included in appendix (Table G.1).

To check the plausibility of the generated molecules, we then run a series of quality filters from Brown et al. [5], which aim to detect those which are *"potentially unstable, reactive, laborious to synthesize, or simply unpleasant"*. Of our valid generated molecules, we find 71.3% pass the quality filters, nearly the same success rate as the test set molecules themselves, 72.2%; if we were to normalize as in Table 1 of Bradshaw et al. [4], our performance of $\approx 98.7\%$ outperforms nearly all approaches considered. Figure 1 shows example generations from the model trained on the ChEMBL, 12 out of

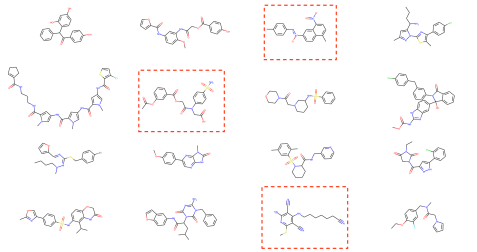

Figure 1: Example generated molecules from the ChEMBL model; red indicates failing the quality filters.

Table 6: Comparing with the conditional VAE model of Kang and Cho [14] on the conditional molecule generation task.

| Target | Kang & Cho | Ours |
|---|---|---|
| MolWt=250 | 250.3±6.7 | 253.8 ±11.8 |
| MolWt = 350 | 349.6±7.3 | 351.7 ±12.5 |
| MolWt = 450 | 449.6±8.9 | 450.9±13.2 |
| Logp = 1.5 | 1.539±0.301 | 1.571±0.371 |
| Logp = 3 | 2.984±0.295 | 3.034± 0.348 |
| Logp = 4.5 | 4.350±0.309 | 4.499±0.338 |
| QED= 0.5 | 0.527±0.094 | 0.502±0.079 |
| QED = 0.7 | 0.719±0.088 | 0.691 ±0.063 |
| QED = 0.9 | 0.840± 0.070 | 0.882± 0.044 |

16 generated molecules passed the quality filter. In Fig. K.9 in appendix we provide an example of ten generations our QM9 model produces when we condition on a given target property vector. Among the ten generated molecules we have nine distinct ones. Five of these (the boxed ones) have never been seen in the training set. QM9 does not represent any real molecule distribution — we do note that all the molecules we show have 9 heavy atoms, consistent with the training dataset. Our model does not simply rank and propose training instances for a given property: it successfully generates novel molecules.

### 3.3 Testing against a strong baseline: Data augmentation -based sampling

Here we explore two additional data augmentation -based approaches to sampling of learning instances for the molecule generation datasets. The first of these is RAML [18], which maximizes a conditional log probability of the augmented versions of the training instances. Given a training instance $(\mathbf{x}^*, \mathbf{y}^*)$, it samples from a distribution $q(\mathbf{x}|\mathbf{x}^*; \tau)$ which is implicitly defined by an appropriate augmentation/perturbation strategy from the training instance $\mathbf{x}^*$. It was designed for text tasks such as machine translation, where for example the training instances correspond to sentences in a source and target language. In that setting an effective perturbation strategy is simply an edit-distance-based modification of the $\mathbf{x}^*$ instance. Such an approach will work well when the augmentation strategy produces instances $\mathbf{x} \sim q(\mathbf{x}|\mathbf{x}^*; \tau)$ which will exhibit a property $\mathbf{y}$ that does not significantly stray away from from $\mathbf{y}^*$ — that is, $f(\mathbf{x}) \approx f(\mathbf{x}^*)$ when $\mathbf{x}$ and $\mathbf{x}^*$ are close in edit distance. We will evaluate the ability of edit-distance-based augmentation to generate sequences whose properties are close to those of the original training sequence $\mathbf{x}^*$, with edit-distance-based RAML as a baseline. In addition, since we can evaluate the properties of the augmented sequences we will do so and add the resulting $(\mathbf{x}, \mathbf{y} = f(\mathbf{x}))$ instances in the an extended training set, in what can be seen as a standard data augmentation strategy for an additional baseline. Note this is only possible because in this particular setting we are able to evaluate our ground truth function $f$; in general one cannot expect this to be available.

We also tested a REINFORCE approach where we actually use a score function gradient estimator (Eq. 2), with warm-start from a model trained with maximum likelihood. This is uncompetitive with any of the other baselines; we discuss these RL results in Appendix I, and Tables I.4 and I.5. Moreover, we also compare with other existing models that purely designed for conditional molecule generation. Most other work for molecule generation cannot do so in one step, instead using the model as part of an iterative optimization procedure (e.g. RL or Bayesian optimization). The most competitive model we are aware of is Kang and Cho [14], which can indeed do direct conditional generation. In Table 6 we compare our results using the ChEMBL-trained model on the task considered in their paper, generating conditioned on a single target property. Despite our model not being tailored for this task, we perform similarly well or better in terms of property accuracy, and furthermore, we do so *far faster* — their model employs a beam search decoder averaging 4.5 *seconds* per molecule, with ours requiring 6 *milliseconds*.

**Edit distance on SMILES**  We first study the appropriateness of the edit-distance-based augmentations, used in RAML, in the sequence problems we examine. Given a sampled SMILES string from the training set we do $m$ edit-distance-based perturbations of it by removing, inserting or swapping $m$ number of the characters. We sample randomly 100 SMILES string from the training set and

| $m$ | validity | unicity | MSE |
|---|---|---|---|
| One | 0.265 | 0.322 | 194.945 |
| Two | 0.095s | 0.421 | 468.556 |
| Three | 0.046 | 0.393 | 725.128 |
| Four | 0.0276 | 0.422 | 985.451 |
| Five | 0.0204 | 0.480 | 1496.023 |
| Six | 0 | - | - |

Table 7: Edit distance augmentation evaluation on QM9 dataset

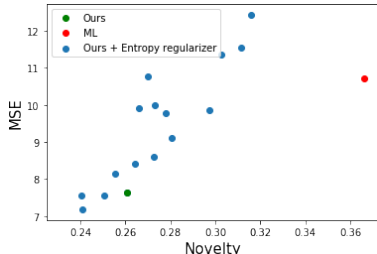

Figure 2: Effect of the entropy on the generated sequences from the validation set.

| | QM9: MSE | | | | | | | | |
|---|---|---|---|---|---|---|---|---|---|
| Model | # rotatable bonds | # aromatic ring | logP | QED | TPSA | bertz | molecule weight | fluorine count | # rings |
| Classic data augmentation | 0.0584 | 0.0107 | 0.0631 | 0.0017 | 7.7358 | 133.6868 | 6.4299 | 0.0009 | 0.0755 |
| RAML-like data augmentation | 0.9666 | 0.0876 | 0.5991 | 0.0071 | 181.7502 | 2677.7330 | 2031.7948 | 0.0182 | 0.5356 |
| Ours + entropy ($\lambda = 0.0008$) | **0.0228** | **0.0007** | **0.0262** | **0.0007** | **6.3374** | **80.2370** | **2.2935** | **0.0006** | **0.0191** |
| | QM9: Correlation coefficient | | | | | | | | |
| Classic data augmentation | 0.976660 | 0.970758 | 0.969238 | 0.853202 | 0.991058 | 0.969615 | 0.914065 | **0.987762** | 0.971779 |
| RAML-like data augmentation | 0.662532 | 0.665283 | 0.724079 | 0.499901 | 0.790727 | 0.554120 | 0.080851 | 0.795729 | 0.817079 |
| Ours + entropy ($\lambda = 0.0008$) | **0.990437** | **0.999046** | **0.987524** | **0.941259** | **0.992992** | **0.983400** | **0.982855** | 0.980721 | **0.994428** |

Table 8: Conditional generation performance for the molecule datasets of the data augmentation based sampling and our entropy regulariser. Due to space constraints, standard deviations are omitted here, but can be found in Appendix Table L.7.

generate for each one of them 1000 augmentations for every $m \in [1, 6]$. We report the validity and uniqueness of the perturbed molecules and the distance (MSE) of their properties from the properties of the original molecule in Table 7.

As we can see in Table 7, the property of a molecule is very sensitive to perturbations. The same holds for the integer expressions as well. Even a one character edit can lead to a drastic property change. This puts into question the use of edit-distance-based augmentations as a means to sample learning instances when we deal with sequences that are sensitive to local perturbations.

**Importance sampling with edit distance proposals** Although our approach restricts the domain of the normalized rewards distribution to allow easy sampling, our reward function Eq. (12) is able to assign reward for any given sequence. Since the edit-distance-based augmentation can propose new sequences, we can use the $q(\mathbf{x}|\mathbf{x}^*, \tau)$ as a proposal distribution and importance sample, with

$$\mathbb{E}_{\bar{R}(\mathbf{x}|\mathbf{y}^*)} \left[\log p_\theta(\mathbf{x}|\mathbf{y}^*)\right] = \mathbb{E}_{q(\mathbf{x}|\mathbf{x}^*, \tau)} \left[ \frac{\bar{R}(\mathbf{x}|\mathbf{y}^*)}{q(\mathbf{x}|\mathbf{x}^*, \tau)} \log p_\theta(\mathbf{x}|\mathbf{y}^*) \right].$$

We follow the same edit distance sampling process as Norouzi et al. [18] and set $\tau = 0.745$ so that if we sample 10 values, the zero edit distance will be sampled with $p = 0.1$. The average $\ell_1$ distance of the properties of the augmented instances from the target property is 4641.0. As a result most of the sequences proposed by $q(\mathbf{x}|\mathbf{x}^*, \tau)$ will have $\bar{R}(\mathbf{x}|\mathbf{y}^*) = 0$ and contribute nothing to learning.

**Data augmentation on SMILES** Even though the edit distance does not preserve properties, we can still evaluate their properties using RDKit; thus, we can pair every augmented instance $\mathbf{x}$ with its evaluated property $f(\mathbf{x})$, and add these couples in the training dataset. We call this baseline "Classic data augmentation". For completeness we will also evaluate the "RAML-like data augmentation" where we pair the augmented instances $\mathbf{x}$ with the property of the original $\mathbf{x}^*$ from which they were produced, even though as we have seen such a pairing is not appropriate.

In Table 4 we give the quality of the generations of the augmentation-based sampling approaches. In terms of validity the two augmentation-based approaches have a lower performance compared to our method in Table 3. For novelty they have considerably higher scores, to be expected since they are actually generating new training data. However, when it comes to the quality of the conditional generations, as measured by MSE in, Table 8, as expected the RAML-based approach performs very poorly, and while "Classic data augmentation" performs acceptably is it still considerably worse than our training method results in Table 5.

### 3.4 Deploying the entropy-based regulariser

Our basic approach achieves a high novelty score on the larger ChEMBL dataset. However this is not the case for the smaller dataset QM9. In an effort to improve the novelty of our method in small data regimes, we introduce the entropy regulariser, from Eq. (10). We evaluate its effect for $\lambda$ in a range of values from 0.0001 to 0.005 for QM9, using the same experimental settings as above. We measured trained models' generation performance on the hold out validation set. Uniqueness stays rather stable, while validity slightly drops as $\lambda$ increases. Fig. 2 shows there is a trade-off between novelty and conditional generation performance: increased novelty comes at the cost of property MSE. In Tables 4 and 8, we include the results of using the entropy-regularized model trained with a $\lambda$ value that improves the novelty while still matching MSE performance of ML training.

## 4 Related work

Neural sequence models [2, 23] are typically trained via maximum likelihood estimation. However, the log-likelihood objective is only measured on the ground truth input-output pairs: such training does not take into account the fact that for a target property, there exist many candidates that are different from the ground truth but still acceptable. To account for such a disadvantage, Norouzi et al. [18] proposed reward-augmented maximum likelihood (RAML) which adds reward-aware perturbations to the target, and Xie et al. [27] applied data noising in language models. Although these works show improvement over pure ML training, they make an implicit assumption that the underlying sequences' properties are relatively stable with respect to small perturbations. This restricts the applicability of such models on other types of sequences than text.

An alternative is a two-stage optimization approach, in which an initial general-purpose model is later fine-tuned for conditional generation tasks. Segler et al. [22] propose using an unconditional RNN model to generate a large body of candidate sequences; goal-directed generation is then achieved by either filtering the generated candidates or fine-tuning the network. Alternatively, autoencoder-based methods map from the discrete sequence data into a continuous space. Conditioning is then done via optimization in the learned continuous latent representation [9, 15, 7], or through direct generation with a semi-supervised VAE [14] or through mutual information regularization [1]. Similar work appeared in conditional text generation [12], where a constraint is added so that the generated text matches a target property by jointly training a VAE with a discriminator.

Instead of using conditional log-likelihood as a surrogate, direct optimization of task reward has been considered a gold standard for goal-oriented sequence generation tasks. Various RL approaches have been applied for sequence generation, such as policy gradient [20] and actor-critic [3]. However, maximizing the expected reward is costly. To overcome such disadvantages, one need to apply various variance reduction techniques [10], warm starting from a pre-trained model, incorporating beam search [26, 6] or a replay buffer during training. More recent works also focused on replacing the sampling distribution with some other complicated distribution which takes into account both the model and reward distribution [8]. However, those techniques are still expensive and hard to train. Our work in this paper aims to develop an easy alternative to optimize the expected reward.

## 5 Conclusion

We present a simple, tractable, and efficient approach to expected reward maximization for goal-oriented discrete structure generation tasks. By sampling directly from the approximate normalized reward distribution, our model eliminates the sample inefficiency faced when using a score function gradient estimator to maximize the expected reward. We present results on two conditional generation tasks, finding significant reductions in conditional generation error across a range of baselines.

One potential concern with this approach would be that reliance on a training dataset could reduce novelty of generations, relative to reinforcement learning algorithms that actively propose new structures during training. This is particularly a concern in a low-data regime: while the entropy-based regularization can help, it may not be enough to help discover (say) entire modes missing from the training dataset. We leave this exploration to future work. However, in some settings the implicit bias of the dataset may be beneficial: if the training data all is drawn from a distribution of "plausible" values, then deviations too far from training examples may be undesirable. For example, in some molecule design settings the reward may be measured by a machine learning property predictor, fit to its own training dataset; if an RL algorithm finds points the property predictor suggests are promising, but are far from the training data, they may be spurious false positives.

## Broader Impact

When a training dataset of suitable examples is available, the methodology introduced by this paper provides an easy approach to learning conditional models. These models otherwise would be fit using an expensive reinforcement learning algorithm, or by less-performant maximum likelihood estimation. Our hope is that learning algorithms that are simpler to tune can help drive adoption of machine learning methods by the broader scientific community, and may help reduce computational and energy requirements for training these models. Furthermore, we are not aware of existing work that trains a single model capable of conditional generation of molecules given such a diverse set of properties; we believe this application and its results will be of independent interest to the computational chemistry community.

## Acknowledgments and Disclosure of Funding

This work was supported by the Alan Turing Institute under the EPSRC grant EP/N510129/1 and the RCSO ISNet within the "Machine learning tools for target molecule design" project.

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
