[Supplementary Material]

## A  Equivalent formulation of the expected reward

$$\mathbb{E}_{p_\theta(\mathbf{x}|\mathbf{y})}[R(\mathbf{x};\mathbf{y})] = \sum_{\mathbf{x}}[p_\theta(\mathbf{x}|\mathbf{y})R(\mathbf{x};\mathbf{y})] \tag{13}$$

$$= \sum_{\mathbf{x}}[\frac{R(\mathbf{x};\mathbf{y})}{\sum_{\mathbf{x}}R(\mathbf{x};\mathbf{y})}p_\theta(\mathbf{x}|\mathbf{y})(\sum_{\mathbf{x}}R(\mathbf{x};\mathbf{y}))]$$

$$= (\sum_{\mathbf{x}}R(\mathbf{x};\mathbf{y}))\sum_{\mathbf{x}}[\frac{R(\mathbf{x};\mathbf{y})}{\sum_{\mathbf{x}}R(\mathbf{x};\mathbf{y})}p_\theta(\mathbf{x}|\mathbf{y})]$$

$$= c(\mathbf{y})\sum_{\mathbf{x}}[\bar{R}(\mathbf{x}|\mathbf{y})p_\theta(\mathbf{x}|\mathbf{y})]$$

$$= c(\mathbf{y})\mathbb{E}_{\bar{R}(\mathbf{x}|\mathbf{y})}p_\theta(\mathbf{x}|\mathbf{y})$$

where $c(\mathbf{y}) = \sum_{\mathbf{x}}R(\mathbf{x};\mathbf{y})$, $\bar{R}(\mathbf{x}|\mathbf{y}) = R(\mathbf{x};\mathbf{y})/c(\mathbf{y})$

## B  Approximating the normalized reward

As our final objective presented in Equation (5), we need to draw samples from the normalized reward distribution $\bar{R}(\mathbf{x}|\mathbf{y}_i)$. We proposed an distribution $p(i,j)$ which is defined on training data to approximate such distribution. Here we are going to discuss the derivation of such approximation and it's effect on the final learning.

$$\mathbb{E}_{\bar{R}(\mathbf{x}|\mathbf{y}_i)}[\log p_\theta(\mathbf{x}|\mathbf{y}_i)] \approx \sum_{j=1}^{N}[\bar{R}(\mathbf{x}_j|\mathbf{y}_i)\log p_\theta(\mathbf{x}_j|\mathbf{y}_i)]$$

$$= \sum_{j=1}^{N}[\frac{R(\mathbf{x}_j;\mathbf{y}_i)}{\sum_{\mathbf{x}}R(\mathbf{x};\mathbf{y}_i)}\log p_\theta(\mathbf{x}_j|\mathbf{y}_i)]$$

$$= \sum_{j=1}^{N}[\frac{R(\mathbf{x}_j;\mathbf{y}_i)}{\sum_{\mathbf{x}}R(\mathbf{x};\mathbf{y}_i)}\frac{\sum_{j=1}^{N}R(\mathbf{x}_j;\mathbf{y}_i)}{\sum_{j=1}^{N}R(\mathbf{x}_j;\mathbf{y}_i)}\log p_\theta(\mathbf{x}_j|\mathbf{y}_i)]$$

$$= \sum_{j=1}^{N}[\frac{\sum_{j=1}^{N}R(\mathbf{x}_j;\mathbf{y}_i)}{\sum_{\mathbf{x}}R(\mathbf{x};\mathbf{y}_i)}\frac{R(\mathbf{x}_j;\mathbf{y}_i)}{\sum_{j=1}^{N}R(\mathbf{x}_j;\mathbf{y}_i)}\log p_\theta(\mathbf{x}_j;\mathbf{y}_i)]$$

$$= \frac{\sum_{j=1}^{N}R(\mathbf{x}_j;\mathbf{y}_i)}{\sum_{\mathbf{x}}R(\mathbf{x};\mathbf{y}_i)}\sum_{j=1}^{N}[\frac{R(\mathbf{x}_j;\mathbf{y}_i)}{\sum_{j=1}^{N}R(\mathbf{x}_j;\mathbf{y}_i)}\log p_\theta(\mathbf{x}_j|\mathbf{y}_i)]$$

$$= \frac{\sum_{j=1}^{N}R(\mathbf{x}_j;\mathbf{y}_i)}{\sum_{\mathbf{x}}R(\mathbf{x};\mathbf{y}_i)}\mathbb{E}_{p(j|i)}[\log p_\theta(\mathbf{x}_j|\mathbf{y}_i)]$$

$$\approx \mathbb{E}_{p(j|i)}[\log p_\theta(\mathbf{x}_j|\mathbf{y}_i)] \tag{14}$$

where $p(j|i)$ is defined as

$$p(j|i) = \frac{R(\mathbf{x}_j;\mathbf{y}_i)}{\sum_{j=1}^{N}R(\mathbf{x}_j;\mathbf{y}_i)}. \tag{15}$$

$$\bar{R}(\mathbf{x}|\mathbf{y}_i) = \frac{R(\mathbf{x}_j;\mathbf{y}_i)}{\sum_{\mathbf{x}}R(\mathbf{x};\mathbf{y}_i)} \tag{16}$$

Note that the original normalized distribution $\bar{R}(\mathbf{x}|\mathbf{y}_i)$ is defined on all possible $\mathbf{x}$ and normalized by $\sum_{\mathbf{x}}R(\mathbf{x};\mathbf{y}_i)$, which is the summation over all possible $\mathbf{x}$, while the distribution $p(i,j)$ is defined only on the training set and therefore normalized by $\sum_{j=1}^{N}R(\mathbf{x}_j;\mathbf{y}_i)$ which is summation over all $\mathbf{x}$ that are in the training set. The last approximation in equation ?? is off by a scalar multiplication factor $\sum_{j=1}^{N}R(\mathbf{x}_j;\mathbf{y}_i)/\sum_{\mathbf{x}}R(\mathbf{x};\mathbf{y}_i)$.

However, this factor does not dramatically affect outcomes for two reasons. First, its value *independent of the value of* $\mathbf{x}$, depending only on $\mathbf{y}_i$. Any bias which is introduced would come into play only in the evaluation of equation 6, where entries from different $\mathbf{y}$ values may effectively be assigned different weights. So while this may affect unconditional generations from the model, and means that some $\mathbf{y}$ values may be considered "more important" during training, it should not directly bias *conditional* generation for a fixed $\mathbf{y}$.

Second, this can be thought of as a ratio of two expectations; the numerator w.r.t. a uniform distribution on x and the denominator w.r.t. the empirical data distribution $\tilde{p}(\mathbf{x})$. For one of the empirical examples (on QM9), this ratio has an expected value of 1.0, since the dataset itself is constructed by enumeration of all molecules with up to 9 heavy atoms and thus also follows a uniform distribution over the domain.

For the other examples, if the training data well represents the underlying conditional distribution, that scaling factor should be almost equal for all y in the training set and dropping this term would not affect the optimization. This is because both terms correspond to estimates of expected reward — they only differ if values of $\mathbf{x}$ which have high probability under the uniform distribution (and low probability under the data distribution) also *have high reward for a given target* $\mathbf{y}_i$. This factor then should not vary largely unless significant modes of x are missing from the training data for some values of $\mathbf{y}_i$, but not for others.

## C   Gradient of the entropy

The gradient of the exact entropy term can be calculated as following using log derivative trick:

$$\nabla_\theta H(p_\theta(\mathbf{x}|\mathbf{y})) = -\nabla_\theta[\sum_{\mathbf{x}} p_\theta(\mathbf{x}|\mathbf{y}) \log p_\theta(\mathbf{x}|\mathbf{y})] \tag{17}$$

$$= -\sum_{\mathbf{x}} \nabla_\theta[p_\theta(\mathbf{x}|\mathbf{y}) \log p_\theta(\mathbf{x}|\mathbf{y})]$$

$$= -\sum_{\mathbf{x}}[p_\theta(\mathbf{x}|\mathbf{y})\nabla \log p_\theta(\mathbf{x}|\mathbf{y}) + \log p_\theta(\mathbf{x}|\mathbf{y})\nabla_\theta p_\theta(\mathbf{x}|\mathbf{y})]$$

$$= -\sum_{\mathbf{x}}(1 + \log p_\theta(\mathbf{x}|\mathbf{y}))p_\theta(\mathbf{x}|\mathbf{y})\nabla_\theta \log p_\theta(\mathbf{x}|\mathbf{y})$$

$$= -\mathbb{E}_{p_\theta(\mathbf{x}|\mathbf{y})}[(1 + \log p_\theta(\mathbf{x}|\mathbf{y}))\nabla_\theta \log p_\theta(\mathbf{x}|\mathbf{y})]$$

## D   Description of the model structure and experiments setup

**Model structure**   To model the conditional distribution $p_\theta(\mathbf{x}|\mathbf{y})$, we modified the Segler et al. [22], Brown et al. [5] sequence model which was initially designed to learn $p_\theta(\mathbf{x})$. The pipeline of the model is presented in figure **??**. We set a maximum length of the sequence $T = 100$. By adding a start and stop token, we represent each sequence with a $T + 2$ length vector, each element of which is an index in the dictionary. We use zero padding whenever it is necessary. Each element of the $T + 2$ vector is embedded to a $h$-dimensional vector (h=512) through an embedding layer and concatenate with its $c$-dimensional property vector. Thus each sequence/molecule is represented with a $(T + 2) \times (h + c)$ matrix. We feed this matrix to an LSTM with three hidden layers with hidden state dimension 512. The output of LSTM last hidden layers is then feed to a linear layer to generates the resulting sequence which is given by a $(T + 2) \times D$ matrix where $D$ refers to the dictionary size that used to describe the sequences.

**experimental setups**   We set the batch size to 20, maximum epoch number to 100 and learning rate of the Adam optimizer to 0.001. We early stop when the validation set performance decreases by a factor of two over the best validation performance obtained so far. Since our learning task is generating sequences that exhibit a given set of desired properties, $\mathbf{y}$, i.e. the properties over which we do the conditioning, we define the validation set performance as the error between the set of properties $\mathbf{y}'$ that the sequence we generate by deterministic decoding from the learned model exhibits, and the desired set properties $\mathbf{y}$ over which we conditioned the generation.

**Datasets**   For the QM9 dataset, we split it into a train, a validation, and a test set with 113k, 10k, an 10k instances, respectively. For the ChEMBL dataset [17], we particularly consider the subset of

Figure D.3: Model pipeline

1600k molecules used for benchmarking by Brown et al. [5]. This dataset is divided in a training set, a validation set and a test set with roughly 1273k, 79k, and 238k instances respectively.

**Hyper parameter setting** The value of $\lambda$ and $\epsilon$ in equation 12 are set based on the statistics of $\ell_1(f(\mathbf{x}_j), \mathbf{y}_i)$. Our goal is to have a decent number of suggested $\mathbf{x}$'s that have a property vector that is within $\ell_1$ distance of $\epsilon$ from the desired property vector: a simple heuristic is to choose them such that, if we plan at train time to draw $K$ samples from $p(\cdot|i)$, we see the original paired values $\mathbf{x}_j$ with probability roughly $1/K$; appropriate values of $\epsilon$ can then be selected by inspecting the dataset. For example, when we condition on 9 properties in QM9, while sampling $K = 10$ values of $\mathbf{x}_j$ for each $\mathbf{y}_i$ when evaluating the loss, we set $\epsilon = 0.3$ and $\lambda = 1$: under these values for any given $\mathbf{y}$ from the training set we have a minimum of one and a maximum of 168 suggested $\mathbf{x}$'s, with an average of 13. Similarly for ChEMBL dataset, we set $\epsilon = 0.4$. If we condition on a single, smooth, property, such as LogP, we set $\epsilon = 5 \times 10^{-5}$ and $\lambda = 10^5$, since in that case we can find many molecules that have a practically identical properties.

## E  Python expressions dataset generation

. We generate synthetic training data by sampling form the probabilistic context free grammar presented in Listing 1. We filter the generated expressions to keep only those that evaluate to a value in the range $(-1000, 1000)$, and where the overall length of the expression is at most 30 characters. We generate 500,000 samples and after removing duplicates are left with 308,722 unique (expression, value) pairs. Out of these we set aside 20k pairs as a validation set and an additional 10k as a test set. To learn the conditional generative model $p_\theta(\mathbf{x}|\mathbf{y})$, we rescale the input values $\mathbf{y}$ by a factor of 1000, to ensure that the inputs to the LSTM are in the interval $(-1, 1)$.

## F  Additional details regarding the SMILES data augmentation process

The augmentation on SMILES is done as following: for each SMILES string in the training data, we use the RAML [18] defined edit distance sampling process, setting the $\tau = 0.745$, sampling an edit distance and then applying a transformation on the SMILES string. We keep sampling until we get 10 valid molecules from each SMILES string in the training set. After filtering out replicated ones, we are left with 733k instances, which is 6 times larger than the original training set size. We then pair the augmented smiles with either the properties of the original matching molecule (i.e. the property of the original SMILES), for the RAML-style importance sampling approach, or with the properties that obtained from the RDKit chemical software for the pure data augmentation approach.

## G  More experimental results

In Table **??** we provide conditional generation performance of our model and ML baseline in terms of total MSE and negative log-likelihood computed over the test set.

|  | QM9 | | ChEMBL | |
| --- | --- | --- | --- | --- |
|  | total MSE | $-\log p(\mathbf{x}|\mathbf{y})$ | total MSE | $-\log p(\mathbf{x}|\mathbf{y})$ |
| ML | $10.7237\pm 0.4915$ | **0.2213440** | **204.4400±4.2766** | **0.2494** |
| Ours | **7.6398± 0.2891** | 0.2357489 | 302.5634±5.6231 | 0.2569 |

Table G.1: Conditional generation performance

Table **??** presents the results on full Guacamol ChEMBL test set in terms of generation and conditional generation performance.

| Model | MSE per-property | | | | | | | | | total MSE |
| --- | --- | --- | --- | --- | --- | --- | --- | --- | --- | --- |
|  | On larger size sequence dataset: ChEMBL | | | | | | | | | |
|  | # rotatable bonds | # aromatic ring | logP | QED | TPSA | bertz | molecule weight | fluorine count | # rings | |
| ML | **0.1567** | 0.0376 | 0.1448 | 0.0051 | **27.2466** | **1652.6992** | **103.3155** | 0.0217 | 0.0264 | **202.4087** |
| Ours | 0.1589 | **0.0272** | **0.1331** | **0.0046** | 34.9984 | 2534.8578 | 177.042 | **0.0072** | **0.0190** | 290.2351 |

Table G.2: Conditional generation performance on ChEMBL full testset

To test if the model is able to generate diverse molecules for a given target property, we sampled 10 samples for each $\mathbf{y}$ in the testset of QM9 and measured the validity, uniqueness, and novelty of the generated molecules. The result is presented in Table **??**.

| Model | MSE per-property | | | | | | | | | Validity | Uniqueness | Novelty |
| --- | --- | --- | --- | --- | --- | --- | --- | --- | --- | --- | --- | --- |
|  | On small size sequence dataset: QM9 | | | | | | | | | | | |
|  | # rotatable bonds | # aromatic ring | logP | QED | TPSA | bertz | molecule weight | fluorine count | # rings | | | |
| ML | 0.00918 | 0.00065 | 0.010993 | 0.000322 | 2.82087 | **19.56099** | 0.78806 | 0.00247 | 0.011853 | 0.96420 | **0.558276** | **0.645831** |
| Ours | **0.00400** | **0.00023** | **0.00553** | **0.00012** | **1.08904** | 23.19567 | **0.36655** | **0.00010** | **0.00232** | **0.98781** | 0.512173 | 0.61363 |
|  | Correlation coefficient | | | | | | | | | | | |
| ML | 0.996233 | 0.998001 | 0.994686 | 0.971792 | 0.996827 | 0.995805 | 0.993406 | 0.972188 | **0.995973** | - | - | - |
| Ours | **0.998357** | **0.999288** | **0.997274** | **0.988833** | **0.998773** | **0.99503** | **0.996947** | **0.998834** | 0.999213 | - | - | - |

Table G.3: Generation and conditional generation performance of our model when we sample 10 molecules per property vector in the testset. To calculate the MSE and correlation coefficient, we use mean of the 10 sampled molecules property as $\mathbf{y}'$.

## H   KL divergence as objective

We want to recover the true underlying data distribution $\tilde{p}(\mathbf{x}|\mathbf{y})$ as accurately as possible from the training data that is observed. The KL divergence between the model $p_\theta(\mathbf{x}|\mathbf{y})$ and true data distribution $\tilde{p}(\mathbf{x}|\mathbf{y})$ is given by

$$D_{KL}[p_\theta(\mathbf{x}|\mathbf{y})||\tilde{p}(\mathbf{x}|\mathbf{y})] \tag{18}$$
$$= \mathbb{E}_{p_\theta(\mathbf{x}|\mathbf{y})} \log p_\theta(\mathbf{x}|\mathbf{y}) - \mathbb{E}_{p_\theta(\mathbf{x}|\mathbf{y})} \log \tilde{p}(\mathbf{x}|\mathbf{y})$$
$$= -H(p_\theta(\mathbf{x}|\mathbf{y})) - \mathbb{E}_{p_\theta(\mathbf{x}|\mathbf{y})} \log \tilde{p}(\mathbf{x}|\mathbf{y}).$$

If we minimize KL divergence in this direction,

$$\min_\theta D_{KL}[p_\theta(\mathbf{x}|\mathbf{y})||\tilde{p}(\mathbf{x}|\mathbf{y})] \approx \tag{19}$$
$$\max_\theta \mathbb{E}_{p_\theta(\mathbf{x}|\mathbf{y})} \log \tilde{p}(\mathbf{x}|\mathbf{y}) + H(p_\theta(\mathbf{x}|\mathbf{y})),$$

and assume a non-parametric form approximation of the true distribution $\tilde{p}(\mathbf{x}|\mathbf{y}) \approx \frac{\exp(R(\hat{\mathbf{x}},\mathbf{y}_i))}{\sum_{\hat{\mathbf{x}}} \exp R(\hat{\mathbf{x}},\mathbf{y}_i)}$, where $R(\hat{\mathbf{x}}, \mathbf{y}_i)$ refers to some reward function, we get exactly the expected reward objective with maximum entropy regularizer.

If we take KL in the opposite direction

$$D_{KL}[\tilde{p}(\mathbf{x}|\mathbf{y})||p_\theta(\mathbf{x}|\mathbf{y})]$$
$$= \mathbb{E}_{\tilde{p}(\mathbf{x}|\mathbf{y})} \log \tilde{p}(\mathbf{x}|\mathbf{y}) - \mathbb{E}_{\tilde{p}(\mathbf{x}|\mathbf{y})} \log p_\theta(\mathbf{x}|\mathbf{y})$$
$$= -H(\tilde{p}(\mathbf{x}|\mathbf{y})) - \mathbb{E}_{\tilde{p}(\mathbf{x}|\mathbf{y})} \log p_\theta(\mathbf{x}|\mathbf{y}) \tag{20}$$

we have

$$\min D_{KL}[\tilde{p}(\mathbf{x}|\mathbf{y})||p_\theta(\mathbf{x}|\mathbf{y})] \approx \max_\theta \mathbb{E}_{\tilde{p}(\mathbf{x}|\mathbf{y})} \log p_\theta(\mathbf{x}|\mathbf{y}). \tag{21}$$

As the expectation is taken over the true data distribution, one can empirically evaluate it on the training data pairs. This is equivalent of assuming that $\tilde{p}(\mathbf{x}|\mathbf{y}) \approx \delta(\mathbf{x}|\mathbf{y})$ and doing maximum log likelihood training on the training set. Even though both KL have a hypothetical minimum at $p_\theta(\mathbf{x}|\mathbf{y}) = \tilde{p}(\mathbf{x}|\mathbf{y})$, they do not achieve the same solution unless the model has enough learning capability. $D_{KL}[p_\theta(\mathbf{x}|\mathbf{y})||\tilde{p}(\mathbf{x}|\mathbf{y})]$ encourages $p_\theta(\mathbf{x}|\mathbf{y})$ to put its mass mainly on the region where true data distribution $\tilde{p}(\mathbf{x}|\mathbf{y})$ has concentrated mass, while the $D_{KL}[\tilde{p}(\mathbf{x}|\mathbf{y})||p_\theta(\mathbf{x}|\mathbf{y})]$ pushes $p_\theta(\mathbf{x}|\mathbf{y})$ to learn to cover all the region that $\tilde{p}(\mathbf{x}|\mathbf{y})$ has its mass on [16, 13].

# I  RL baseline

Our objective is to maximize the expected reward:

$$\mathcal{J} = \mathbb{E}_{\tilde{p}(\mathbf{y})}\mathbb{E}_{p_\theta(\mathbf{x}|\mathbf{y})}[R(\mathbf{x};\mathbf{y})]. \tag{22}$$

where $R(\mathbf{x};\mathbf{y}) = \exp\{-\lambda d(f(\mathbf{x}),\mathbf{y})\}$. Using the data distribution to approximate expectations over $\tilde{p}(\mathbf{y})$, we have:

$$\mathcal{J} \approx \frac{1}{N}\sum_{i=1}^{N}\mathbb{E}_{p_\theta(\mathbf{x}|\mathbf{y}_i)}[R(\mathbf{x};\mathbf{y}_i)]. \tag{23}$$

where $\mathbf{y}_i$ is sampled from training data.

Note that in our case, the model $p_\theta(\mathbf{x}|\mathbf{y})$ defines a distribution over discrete random variables and reward depends on non-differentiable oracle function $f$ that return feedback on the discrete sequence $\mathbf{x}$ that is sampled from the model $p_\theta(\mathbf{x}|\mathbf{y})$. Therefore, we can not directly differentiate the $\hat{\mathcal{J}}$ with respect to the model parameter $\theta$. One way to apply gradient based optimization in this case is a to use score-function estimators of the gradient:

$$\nabla_\theta \mathcal{J} \approx \frac{1}{N}\sum_{i=1}^{N}\mathbb{E}_{p_\theta(\mathbf{x}|\mathbf{y}_i)}[R(\mathbf{x};\mathbf{y}_i)\nabla_\theta \log p_\theta(\mathbf{x}|\mathbf{y}_i)]$$

$$\approx \frac{1}{NM}\sum_{i=1}^{N}\sum_{j=1}^{M}[R(\mathbf{x}_j;\mathbf{y}_i)\nabla_\theta \log p_\theta(\mathbf{x}_j|\mathbf{y}_i)]. \tag{24}$$

The score-function gradient estimators have high variance. Besides, in the beginning, the output of the model mostly corresponding to invalid sequences. Therefore, we use to initialize our model from a pre-trained model as a warm-start. The pre-trained model is obtained by maximizing the log-likelihood of the training data.

In the following experiment, we train the same model with the maximum log-likelihood objective for six epochs to obtain the pre-trained model for warm start. We set sample size $M = 30$, mini-batch size = 20. We set the temperature parameter $\lambda = 0.5$. At the early stage of training, since the model is not perfect, the invalid samples proposed by the model are discarded. Note that such training is very time-consuming because during training at each mini-batch, firstly, we need to sample from the model by involves unrolling the RNN which is pretty slow when we have long sequences. Secondly, for each sampled sequence, to get the property, we need to send it to some oracle function, in this case, RDKit, which is normally implemented in CPU, this requires frequent communication between CPU and GPU which greatly increases computation time.

| Model | QM9 | | | |
| | Validity | Unicity | Novelty | Training time per epoch (hour) |
|---|---|---|---|---|
| ML | 0.9619 | **0.9667** | 0.3660 | 0.19 |
| Ours | **0.9886** | 0.9629 | 0.2605 | 0.56 |
| RL+ warm start | 0.4013 | 0.8425 | **0.8497** | 3.05 |

Table I.4: Molecule generation quality of the data augmentation sampling strategies and our entropy regulariser

| | | | | | QM9: MSE | | | | |
|---|---|---|---|---|---|---|---|---|---|
| Model | # rotatable bonds | # aromatic ring | logP | QED | TPSA | bertz | molecule weight | fluorine count | # rings |
| ML | 0.0468±0.0014 | 0.0014±0.0003 | 0.0390±0.0013 | 0.0010±0.0000 | 11.1772±0.3129 | 80.7725±4.4282 | 4.4251±0.3450 | 0.0023±0.0012 | 0.0484±0.0034 |
| Ours | **0.0166±0.0009** | **0.0005±0.0005** | **0.0184±0.0010** | **0.0004±0.0000** | **3.8585±0.1637** | **63.6678±2.5520** | **1.1835±0.1421** | **0.0004±0.0003** | **0.0120±0.0027** |
| RL+ warm start | 0.3711±0.0149 | 0.0359±0.0030 | 0.5285±0.0141 | 0.0102±0.0002 | 206.6262±3.4869 | 1023.2935±24.5836 | 53.1911±2.2178 | 0.0183±0.0031 | 0.4260±0.0119 |
| | | | | | QM9: Correlation coefficient | | | | |
| ML | 0.980881 | 0.994366 | 0.980527 | 0.906267 | 0.987089 | **0.984265** | 0.965101 | 0.978346 | 0.981742 |
| Ours | **0.993745** | **0.997184** | **0.990115** | **0.963365** | **0.995382** | 0.984006 | **0.988702** | **1.000000** | **0.994824** |
| RL+ warm start | 0.851715 | 0.840686 | 0.830349 | 0.460100 | 0.904118 | 0.827304 | 0.714198 | 0.760453 | 0.899375 |

Table I.5: Conditional generation performance for the molecules datasets

Note that the computational cost for sampling from RNN and frequent communication between CPU and GPU to evaluate the properties of the sampled molecules, do not allow us to use bigger sample size. With sample size 20, after relaxing the early stopping criteria, the training still exist with training loss increases more than 10 times the minimum training loss been obtained so far.

## J  Different ways to approximate the Entropy term

The entropy of $p_\theta(\mathbf{x}|\mathbf{y}) = \prod_{t=1}^{T} p_\theta(\mathbf{x}_t|\mathbf{x}_{1:t-1}, \mathbf{y})$ is given by:

$$H[p_\theta(\mathbf{x}|\mathbf{y})] = -E_{p_\theta(\mathbf{x}|\mathbf{y})}[\log p_\theta(\mathbf{x}|\mathbf{y})] \tag{25}$$

$$= -E_{p_\theta(\mathbf{x}_{1:t}|\mathbf{y})}\left[\sum_{t=1}^{T} \log p_\theta(\mathbf{x}_t|\mathbf{x}_{1:t-1}, \underline{\mathbf{y}})\right].$$

A naïve Monte Carlo estimation involves sampling trajectories $\mathbf{x}$, given $\mathbf{y}$, and then evaluating the log probabilities. We call this approximation ***Estimator A***:

$$\hat{H}_{MC} = -\frac{1}{S}\sum_{s=1}^{S}\sum_{t=1}^{T} \log p_\theta(\mathbf{x}_t^s|\mathbf{x}_{1:t-1}^s, \mathbf{y}) \tag{26}$$

for $\mathbf{x}_t^s \sim p_\theta(\mathbf{x}|\mathbf{y})$.

The alternative way of approximating the entropy involves decomposing this into a sequence of other entropies. In this way, we have

$$H[p_\theta(\mathbf{x}|\mathbf{y})] = H[p_\theta(\mathbf{x}_1|\mathbf{y})] + \sum_{t=2}^{T} E_{p_\theta(\mathbf{x}_{1:t-1}|\mathbf{y})}\left[H[p_\theta(\mathbf{x}_t|\mathbf{x}_{1:t-1}, \mathbf{y})]\right] \tag{27}$$

Since the entropy for each individual $\mathbf{x}_t$ is cheap enough to compute directly in closed form, we can do so and just use sampling in order to generate the values we condition on at each step. We call this approximation ***Estimator B***:

$$H[p_\theta(\mathbf{x}_1|\mathbf{y})] + \sum_{t=2}^{T} E_{p_\theta(\mathbf{x}_{1:t-1}|\mathbf{y})}\left[H[p_\theta(\mathbf{x}_t|\mathbf{x}_{1:t-1}, \mathbf{y})]\right] \approx H[p_\theta(\mathbf{x}_1|\mathbf{y})] + \sum_{t=2}^{T}\sum_{s=1}^{S}\frac{1}{S}H[p_\theta(\mathbf{x}_t|\mathbf{x}_{1:t-1}^s, \mathbf{y})] \tag{28}$$

We randomly sample a $\mathbf{y}$ from the test set and calculated the entropy of $p_\theta(\mathbf{x}|\mathbf{y})$ using above two estimators, with different sample size. We show the histogram of the resulting entropy values over 15 trials on a trained and a random model in figures **??** and **??** respectively. As figure **??** and **??** show, the estimator B is rather stable and has less variance than estimator A, as expected. Therefore, from now on, we use estimator B, which is the Monte Carlo approximation given in the equation (**??**), as a gold standard reference to compare other estimators against. Unfortunately, using the estimator B involves sampling from the model distribution that we want to optimize, so we still have the problems when taking the derivative. An alternative, instead of taking a Monte Carlo approximation of the expectation in front of the each entropy term in equation (**??**), we could do a deterministic greedy decoding by taking the max at each $\mathbf{x}_t$. The below figures **??** and **??** show how the greedy decoding variants of estimator A and B perform against estimator B with Monte Carlo sample size one and 50. For each of the Monte Carlo estimates, we plot a normal distribution showing the mean and standard deviation of the entropy values estimated from the 15 independent trials. Greedy decoding for the values we condition on seems to work reasonably well for approximating the entropy, in both a random model and a trained model setting, which means it is good-enough to use as a regularizer during early stages of training. We also tested the straight through estimator as an alternative to the

Figure J.4: The histogram of the approximated entropy of a fully-trained model $p_{\theta*}(\mathbf{x}|\mathbf{y}_i)$

Figure J.5: The histogram of the entropy of a random model (untrained) $p_{\theta}(\mathbf{x}|\mathbf{y}_i)$

greedy decoding, as it also allows us to take gradients. To get the straight through estimator, instead

Figure J.6: Entropy approximation on the trained model

Figure J.7: Entropy approximation on the random model

of evaluating the RNN on the embedding of a single input, we compute the mean of the embeddings under the distribution.

Figure J.8: Comparing a greedy and straight-through estimator to a Monte Carlo reference

The figure **??** shows, straight through estimator also works reasonably well. In the experiment we use the greedy decoding of the estimator B,

$$
\begin{aligned}
H[p_\theta(\mathbf{x}|\mathbf{y})] =& H[p_\theta(\mathbf{x}_1|\mathbf{y})] + \\
& \sum_{t=2}^{T} E_{p_\theta(\mathbf{x}_{1:t-1}|\mathbf{y})} \left[ H[p_\theta(\mathbf{x}_t|\mathbf{x}_{1:t-1}, \mathbf{y})] \right] \\
\approx& H[p_\theta(\mathbf{x}_1|\mathbf{y})] + \sum_{t=2}^{T} \left[ H[p_\theta(\mathbf{x}_t|\mathbf{x}_{1:t-1}^*, \mathbf{y})] \right],
\end{aligned}
\tag{29}
$$

where $\mathbf{x}_{1:t-1}^*$ is obtained from unrolling the RNN by taking the most probable character at each time step. For this approximation of the entropy the gradient calculation is straightforward. We can calculate the each individual entropy term analytically as our underlying sequence is discrete and finite. Therefore, the gradient calculation of this approximated entropy would be straightforward to implement and cheap in computation time.

## K  Molecule generation baselines

## L  More results

Figure K.9: Molecules generated from a given property value vector. The boxed ones are molecules that have not been seen before.

| Model | # rotatable bonds | # aromatic ring | logP | QED | TPSA | bertz | molecule weight | fluorine count | # rings |
|---|---|---|---|---|---|---|---|---|---|
| | | | | | QM9: MSE | | | | |
| ML | 0.0468±0.0014 | 0.0014±0.0003 | 0.0390±0.0013 | 0.0010±0.0000 | 11.1772±0.3129 | 80.7725±4.4282 | 4.4251±0.3450 | 0.0023±0.0012 | 0.0484±0.0034 |
| Ours | **0.0166±0.0009** | **0.0005±0.0005** | **0.0184±0.0010** | **0.0004±0.0000** | **3.8585±0.1637** | **63.6678±2.5520** | **1.1835±0.1421** | **0.0004±0.0003** | **0.0120±0.0027** |
| | | | | | QM9: Correlation coefficient | | | | |
| ML | 0.980881 | 0.994366 | 0.980527 | 0.906267 | 0.987089 | **0.984265** | 0.965101 | 0.978346 | 0.981742 |
| Ours | **0.993745** | **0.997184** | **0.990115** | **0.963365** | **0.995382** | 0.984006 | **0.988702** | **1.000000** | **0.994824** |
| | | | | | ChEMBL: MSE | | | | |
| ML | **0.1552±0.0104** | 0.0388±0.0028 | 0.1450±0.0025 | 0.0050±0.0001 | **27.6416±0.4204** | **1707.9996±38.8800** | **103.9389±3.1637** | 0.0128±0.0016 | 0.0226±0.0016 |
| Ours | 0.1555±0.0221 | **0.0268±0.0018** | **0.1320±0.0025** | **0.0046±0.0001** | 35.0531±0.4179 | 2512.7421±47.7031 | 174.9301±3.6913 | **0.0074±0.0010** | **0.0191±0.0010** |
| | | | | | CheEMBL: Correlation coefficient | | | | |
| ML | **0.993628** | 0.986184 | 0.977686 | 0.945018 | **0.990576** | **0.993385** | **0.995624** | 0.993952 | 0.993105 |
| Ours | 0.993409 | **0.990111** | **0.979581** | **0.949578** | 0.987775 | 0.990189 | 0.992550 | **0.996583** | **0.994250** |

Table L.6: Conditional generation performance for the molecules datasets

## QM9: MSE

| Model | # rotatable bonds | # aromatic ring | logP | QED | TPSA | bertz | molecule weight | fluorine count | # rings |
|---|---|---|---|---|---|---|---|---|---|
| Classic data augmentation | 0.0584 | 0.0107 | 0.0631 | 0.0017 | 7.7358 | 133.6868 | 6.4299 | 0.0009 | 0.0755 |
| RAML-like data augmentation | 0.9666 | 0.0876 | 0.5991 | 0.0071 | 181.7502 | 2677.7330 | 2031.7948 | 0.0182 | 0.5356 |
| Ours + entropy ($\lambda = 0.0008$) | **0.0228** | **0.0007** | **0.0262** | **0.0007** | **6.3374** | **80.2370** | **2.2935** | **0.0006** | **0.0191** |

## QM9: Correlation coefficient

| Model | # rotatable bonds | # aromatic ring | logP | QED | TPSA | bertz | molecule weight | fluorine count | # rings |
|---|---|---|---|---|---|---|---|---|---|
| Classic data augmentation | 0.976660 | 0.970758 | 0.969238 | 0.853202 | 0.991058 | 0.969615 | 0.914065 | **0.987762** | 0.971779 |
| RAML-like data augmentation | 0.662532 | 0.665283 | 0.724079 | 0.499901 | 0.790727 | 0.554120 | 0.080851 | 0.795729 | 0.817079 |
| Ours + entropy ($\lambda = 0.0008$) | **0.990437** | **0.999046** | **0.987524** | **0.941259** | **0.992992** | **0.983400** | **0.982855** | 0.980721 | **0.994428** |

Table L.7: Conditional generation performance for the molecule datasets of the data augmentation based sampling and our entropy regulariser.