[Reviews · NeurIPS 2020]

Review 1

Summary and Contributions: The paper proposes an approach for generating discrete-structured objects with target properties. The approach is based on an RL formulation that attempts to circumvent the typical high variance of policy gradients. An objective function is introduced that leverages an expectation wrt a normalized reward distribution over examples in the training set in order to train a conditional generative model (policy). Sampling from the model directly is not required during training. Empirical evaluation on two tasks, generation of Python expressions evaluating to a certain target value and generation of molecules matching desired properties, demonstrates favorable performance relative to a model trained with a vanilla maximum-likelihood objective.

Strengths: Relevant: Targeted generation of structured data is an important and active area of research. The proposed method has the potential to make a decent contribution to this area. Well-written methods: The step-by-step description of the method is easy to follow. Sound approach: The high variance of policy gradients in RL is a standard obstacle to its deployment. With some slight re-formulating of the objective, the authors propose a setup that does not require any model sampling during training. Instead, samples are only drawn from the training set using pre-computed probabilities.

Weaknesses: Motivation could be improved: In the introduction, it is stated that a maximum likelihood objective is fundamentally a poor choice for property-conditional generation. This should be expanded upon. Why is this necessarily the case? In the beginning of the related work section, some intuition for this is briefly touched on ("there exist many candidates that are different from the ground truth but still acceptable"), but the related work occurs late in the paper.

Correctness: The methodology appears correct.

Clarity: The paper is well-written and easy to follow. As stated in the weaknesses, I believe a bit more justification of the immediate assumption that RL is necessarily a better choice than maximum likelihood would serve the introduction well.

Relation to Prior Work: The related work section appears adequate.

Reproducibility: Yes

Additional Feedback: I believe the couple suggestions above would improve the presentation. Added after the authors provided their rebuttal: Overall, I'm happy with the author response and that they expanded upon the ML vs RL motivation, which was quite weak in the original text. I will keep my score of 7.


Review 2

Summary and Contributions: This work proposes a reinforcement learning approach to generate sequences of tokens that maximize a desired goal, training on sequence, value pairs. The key contribution is avoiding expensive Monte Carlo steps inherent to policy gradient optimization to maximize expected reward. Instead they utilized a normalized rewards distribution and a path-wise derivative estimator. The normalized rewards distribution is itself approximated from reweighting the training data distribution. In addition, to avoid mode collapse and maximize the sequence diversity of the outcomes, an entropy regularization term is added to the generated sequences.

Strengths: The proposed strategy to bypass Monte Carlo sampling in policy gradient optimization is novel, at least in the molecular design arena (not so sure about other areas). It works to a reasonable degree in the tests shown and slightly overperforms over maximum likelihood while avoiding the computational cost of other RL approaches. The entropy term is effective in bridging with the generative ability of ML approach. The comparison with baselines is appropriate, but not outstanding, and it includes some real-life molecules from ChemBL, synthetic exhaustively enumerated small molecules from QM9, and some comparisons with the Guacamol benchmark set of tasks. There are strong comparisons with other approaches including data augmentation for Maximum Likelihood and Reward Augmented Maximum Likelihood, both of which underperform even the regular Maximum Likelihood approach.

Weaknesses: The improvements over Maximum Likelihood are very moderate and no comparisons are made with more computationally expensive RL approaches (at least on the small QM9 dataset it would be interesting). It would be very interesting to see the performance tradeoff between the proposed approach and the Monte Carlo estimation of the expectation term in Eq 11. One of the promising features of generative algorithms for molecules is their supposed ability to capture a complex statistical distribution of plausible molecules that can be made, paid for, stored in a vial, etc. The approximately 100 million molecules that have been made and the couple of billions that can be confidently said to be makeable are samples from that distribution. It is not clear how much of chemical space is in that manifold. There are many graphs that are formally valid according to valence rules (and their rdkit implementation) that could not exist as molecules because they are not stable. The premise of using generative models for molecular design is sampling natural-looking molecules. Just like generative models for faces, one just needs to look at this to judge whether the model has learned a richer chemistry than the hard-coded rules of RDKit. Very few molecules are shown from what the model produces. It would be great to add samples to the SI and to the accompanying codebase. Of the molecules that are shown, in Figure 1, 5 of them contain primary imine groups that are ¬extremely rare in actual chemistry. The model has not really learned chemical rules (an analogy would be training on celebrity faces and generating celebrity faces with crooked teeth and a mullet over and over). This is particularly interesting because QM9 is an exhaustive enumeration, that is, it contains all the chemically valid molecules (strictly, it’s a superset, it also contains non-chemical molecules). If a molecule with 9 heavy atoms is outside the training data then it is not a true molecule, and just an artifact of the SMILES notation / rdkit valence rules. There are no examples of molecules coming fmor the ChemBL-trained model. Since that is a more natural training dataset (QM9 is a synthetic enumeration) the ability to produce natural-looking samples would be interesting to see. RAML for molecules is interesting but of course it falls onto the same issues, the edit distance in smiles and distance in the manifold of chemical molecules are very different. Mutations from a genetic algorithm might be a more relevant way to edit molecules. (Chem. Sci., 2019,10, 3567-3572)

Correctness: The number of significant digits in Table 5 is confusing? Are they because of the standard deviation? Since the standard deviations are not reported in this table, maybe a summary with fewer digits in the manuscript would be easier to track, and the full numbers be reported in the SI. For most of these chemical properties the decimas digits are meaningless. “atom counter” in line 158 should be atom count? “is generate” in line 65

Clarity: Excellent, very good balance between qualitative explanation and derivations.

Relation to Prior Work: Yes

Reproducibility: Yes

Additional Feedback: The author feedback addresses some of the challenges raised and applies filters to fix plausability issues. Since this is general issue for RL-based approaches, it does not demerit this contribution.


Review 3

Summary and Contributions: The paper addresses the task of conditional generation of discrete structures. The authors introduce an idea of scaling the conditional distribution by the reward R(x|y), which re-weights the samples based on the difference between target molecular features and those of the current sample. The authors test their approach on integer problems and molecule generation. ================================= Added after author response: in the rebuttal the authors explained their motivation more clearly, applicability of other approaches (e.g. RL) and provided a comparison with a competitive baseline by Kang and Cho. They also pointed out the differences with the previous work by Norouzi et al, which was my main concern. This made me change my opinion of the paper, and I am changing my score to 7 (accept).

Strengths: The idea is novel. This approach allows to compute the expected reward across the dataset, rather than samples from the model, avoiding gradient with high variance. The motivation for the approach is sound, and derivations are easy to follow.

Weaknesses: The authors have not made the appropriate comparison to the existing approaches. The authors compared their approach to only two baselines, both of which are weak. The first is a naive baseline of directly training conditional model p(x|y). The second baseline is RAML augmentation, which is clearly underperforming (see Table 8). In the "Related work" authors point out that RAML augmentation is applicable only for text, but for molecules. I would like to see the comparison to other conditional generative models for molecules. The authors mention several of such in "Related work" section, such as Gómez-Bombarelli et al, Kang et al. I have listed more example of conditional models for molecules below. The approach constructing the conditional distribution thorough the scaled distances is novel for the area of generating molecules, but is quite similar to Norouzi et al. The relation to Norouzi et al. should be clearly stated. The authors compared their approach to only one baseline, which is training a conditional model p(x|y). No empirical comparison to other related work is provided. Because of that, it is hard to evaluate the usefulness of the approach.

Correctness: The claims and derivations are correct. In Table 5 it is clear that the features have different scales, and some of them are discrete. The model seems to use the same mean-squared error loss on all the features, the model might optimize for features with larger scale. Was any normalization technique applied to preprocess the features and bring them on the same scale?

Clarity: The paper is clearly written and easy to follow.

Relation to Prior Work: The current paper uses a very similar methodology to Norouzi et al., 2016 (which is mentioned in the paper), where the conditional model is scaled by the normalized exponentiated distances between the samples. Adding the entropy term also follows Norouzi et al. Can authors provide a more detailed explanation how their work differs from Norouzi et al., besides the application area? The authors mention several works for conditional generation of molecules and text. There is a variety of models for conditional generation of molecules that are worth mentioning: Generation of molecules as graphs: Cao et al, 2018 [1], Li et al [2], Li et al. [3], Generation of molecules as SMILES: Lim et al, 2018 [4], Polykovskiy et al. [5], Li et al [2] (used LSTM baseline) [1] MolGAN: An implicit generative model for small molecular graphs. Nicola De Cao, Thomas Kipf, 2018 [2] Learning Deep Generative Models of Graphs. Yujia Li, Oriol Vinyals, Chris Dyer, Razvan Pascanu, Peter Battaglia, 2018 [3] Multi-objective de novo drug design with conditional graph generative model. Yibo Li, Liangren Zhang & Zhenming Liu. Journal of Cheminformatics (2018) [4] Molecular generative model based on conditional variational autoencoder for de novo molecular design Jaechang Lim, Seongok Ryu, Jin Woo Kim & Woo Youn Kim. Journal of Cheminformatics (2018) [5] Daniil Polykovskiy, Alexander Zhebrak, Dmitry Vetrov, Yan Ivanenkov, Vladimir Aladinskiy, Polina Mamoshina, Marine Bozdaganyan, Alexander Aliper, Alex Zhavoronkov, Artur Kadurin. Entangled Conditional Adversarial Autoencoder for de Novo Drug Discovery. Molecular Pharmaceutics, 2018

Reproducibility: Yes

Additional Feedback: It seems that the description for the proposal distribution for q(x| x*, tau) is missing. Can authors clarify how this distribution was constructed?


Review 4

Summary and Contributions: This paper studies the goal-directed generation of structured discrete data problem. It introduces an approach to directly optimize a reinforcement learning objective, that encourages the generation of sequences with specific desired properties. The experiments show that the model is effective to tackle the problem.

Strengths: The proposed idea is sound, although intuitive.

Weaknesses: Presentation and writing of the paper are obscure, making it hard to follow. The contribution is not well- highlighted. The proposed model is simply compared with one baseline, i.e. ML. I am afraid this may not be the first time RL is adopted for structured discrete data generation tasks, i.e. NLP. In this aspect, the contribution of this work may not be enough. The author may need to further highlight their contrition in the rebuttal.

Correctness: The formulation is sound to me.

Clarity: The presentation and writing of the paper are obscure, making it hard to follow. The main contribution is not evident from the current presentation. ===================== I have thoroughly read the author rebuttal. The response addresses my previous concern regarding its motivation and experiments design: Its primary motivation is to beat MLE respect to diverse structured data generation and to save computation cost. That is why they adopt an RL formulation. The proposed RL model is demonstrated to outperform the methods, and it is comparable with other RL method in complex data generation tasks in the Appendix. I have increased my score after the rebuttal. However, I still think the paper needs to be improved in the final version, e.g. detailed motivation statement as in response [R1: Maximum likelihood (ML) VS RL objectives ]. Otherwise, it is hard to understand its motivation from Line 33-44. The logic is not much convictive here.

Relation to Prior Work: The work lacks a sufficient discussion and comparison with other RL works for discrete data. Even they may do not target generation with designed property, they should be briefly discussed to reduce confusions.

Reproducibility: Yes

Additional Feedback: This paper studies the generation of discrete data with specific structured constraints. It motivates to formulate this conditional generation problem in a reinforcement learning setting, namely to learn a stochastic policy p_{\theta}(x | y). It further proposes to sampling from an approximation to the normalized rewards to overcome the requirement of high-variance score-function estimators in existing RL methods. In my aspect, the contribution of this work is not very clear. Is the main contribution/ novelty here, the structured *conditional *generation of discrete data? I mean "to generate discrete data *with specific structured constraints*". Are you the first one to considering this setting? My confusion is that RL methods have been widely adopted in structured discrete data generation problems, e.g. in NLP [1]. However, the paper does not present any discussion on connection or comparison regarding RL methods. The author may need to clearly highlight their motivation in the rebuttal. [1] Fedus, William, Ian Goodfellow, and Andrew M. Dai. "MaskGAN: Better text generation via filling in the_." arXiv preprint arXiv:1801.07736 (2018).

[Author Response · NeurIPS 2020]

We thank all reviewers for their constructive feedback, and provide responses to specific concerns:

**[R1: Maximum likelihood (ML) VS RL objectives ]** The fundamental difference is between "many to one" and "one to many" settings. A ML objective is good choice for prediction tasks where the goal is to match each input to its exact output. This includes models which predict properties from molecules, since each molecule has a well-determined property. However, for the inverse problem the data severely underspecifies the mapping, since any given property combination has many diverse molecules which match; nearly all of these are *not* present in the training data. As the ML objective is only measured on the training pairs, any output that is different from the training data target is penalized. Therefore, a model which generates novel molecules with the correct properties would be penalized by ML training, as it does not produce the exact training pairs; but, it would have a high reward and thus be encouraged by an RL objective.

**[R2: Plausibility of the generated molecules, GA mutation]** Thanks for your insightful comments on plausibility; we include here additional results on molecule quality. Indeed, QM9 does not represent any real molecule distribution — we do note that all the molecules we show have 9 heavy atoms, consistent with the training dataset. Figure 1 shows example generations from the model trained on the ChEMBL. We then run a series of quality filters from Brown et al. [2019], which aim to detect those which are *"potentially unstable, reactive, laborious to synthesize, or simply unpleasant"*. Of our valid generated molecules, we find **71.3% pass the quality filters**, nearly the same success rate as the test set molecules themselves, **72.2%**; if we were to normalize as in Table 1 of Bradshaw et al. [2019], our performance of $\approx$ **98.7**% outperforms nearly all approaches considered. We agree that GA mutations are a more relevant way to edit molecules, but this still suffers from the same fundamental problem that local changes can have large effects on the properties.

**[R3: Molecule generation baselines]** Most other work for molecule generation cannot do so in one step, instead using the model as part of an iterative optimization procedure (e.g. RL or Bayesian optimization). The most competitive model we are aware of is Kang and Cho [2018], which can indeed do direct conditional generation. In Table 2 we compare our results using the ChEMBL-trained model on the task considered in their paper, generating conditioned on a single target property. Despite our model not being tailored for this task, we perform similarly well or better in terms of property accuracy, and furthermore, we do so *far faster* — their model employs a beam search decoder averaging 4.5 *seconds* per molecule, with ours requiring 6 *milliseconds*. Additionally, **their model has only 10% uniqueness** of the generated molecules, in comparison with **81% for us**; a massive increase in diversity.

Figure 1: Example generated molecules from the ChEMBL model; red indicates failing the quality filters.

| Target | Kang & Cho | Ours |
|---|---|---|
| MolWt=250 | 250.3±6.7 | 253.8 ±11.8 |
| MolWt = 350 | 349.6±7.3 | 351.7 ±12.5 |
| MolWt = 450 | 449.6±8.9 | 450.9±13.2 |
| Logp = 1.5 | 1.539±0.301 | 1.571±0.371 |
| Logp = 3 | 2.984±0.295 | 3.034± 0.348 |
| Logp = 4.5 | 4.350±0.309 | 4.499±0.338 |
| QED= 0.5 | 0.527±0.094 | 0.502±0.079 |
| QED = 0.7 | 0.719±0.088 | 0.691 ±0.063 |
| QED = 0.9 | 0.840± 0.070 | 0.882± 0.044 |

Table 2: Comparing with the the strongest baseline on conditional molecule generation task

**[R2, R4: RL baseline]** We actually did run an RL baseline (Eq. 2) on QM9 — we mention this in the main paper (lines 239-241), and the (generally uncompetitive) RL results are in Appendix Tables 1.4 & 1.5. As expected, the novelty of the generated molecules increased greatly as the model was able to simulate novel molecules during training, but the validity and conditional generation performance were far worse. Training such an objective even for QM9 was quite costly (6 times slower than our approach), and for ChEMBL did not converge in a reasonable time. While this might be addressable by more advanced control variate schemes, the main motivation of this paper is to take advantage of the RL objective while avoiding expensive computational costs and unstable training.

**[R3, R4: Contributions, NLP]** As noted conditional generation tasks often occur in NLP, but the setting is usually much more constrained: typically, the "properties" are *individual binary attributes* such as sentiment (positive or negative), or discrete styles ("romantic", "humorous", etc). Conditioning on a small number of binary labels means most methods for style transfer in NLP use auxiliary classifiers as a training signal, and are not easily adapted to larger numbers of real-valued attributes.

**[R3: Relation to RAML]** We discuss the work of Norouzi et al. [2016] in detail in Section 3.3. There are a number of differences, from motivation (we target an RL objective directly) to implementation (we resample potential training sequences). They also do not use the entropy term in training, only to motivate derivations.

[Meta-Review · NeurIPS 2020]

The authors propose an RL-inspired way of fitting a conditional generative model to the training data with the aim of generating discrete structures, such as molecules, satisfying some desired properties. Unlike policy gradients in RL, the proposed algorithm does not require sampling from the model/policy, instead approximating the expectation of interest using the training data reweighted with the normalized rewards. This is done to avoid high gradient variance of policy gradient algorithms. The reviewers liked the novelty of the approach to this important problem. While the experimental results are not spectacular and there were some concerns about missing RL baselines and connections to reward-augmented ML, the author response addressed them in large part. One high level question that needs to be discussed in the paper is whether the proposed method really is a way of optimizing the RL objective as is claimed, or whether it is essentially a weighted maximum likelihood method. For example, unlike a true RL method, the proposed approach seems incapable of discovering truly new structures as it cannot stumble upon them during training because of not sampling from the model. The proposed entropy penalty might just smooth the predictions around the training data rather than lead to discovery of novel structures.